# Automated sequence design of 2D wireframe DNA origami with honeycomb edges

Hyungmin Jun [1,2], Xiao Wang[1,2], William P. Bricker [1] & Mark Bathe [1]*

Wireframe DNA origami has emerged as a powerful approach to fabricating nearly arbitrary 2D and 3D geometries at the nanometer-scale. Complex scaffold and staple routing needed to design wireframe DNA origami objects, however, render fully automated, geometry-based sequence design approaches essential for their synthesis. And wireframe DNA origami structural fidelity can be limited by wireframe edges that are composed only of one or two duplexes. Here we introduce a fully automated computational approach that programs 2D wireframe origami assemblies using honeycomb edges composed of six parallel duplexes. These wireframe assemblies show enhanced structural fidelity from electron microscopy-based measurement of programmed angles compared with identical geometries programmed using dual-duplex edges. Molecular dynamics provides additional theoretical support for the enhanced structural fidelity observed. Application of our top-down sequence design procedure to a variety of complex objects demonstrates its broad utility for programmable 2D nanoscale materials.

[1] Department of Biological Engineering, Massachusetts Institute of Technology, Cambridge, MA 02139, USA. [2] These authors contributed equally: Hyungmin Jun, Xiao Wang *email: mark.bathe@mit.edu

DNA is now well established as a versatile material for nanoscale structural engineering, whereby robust Watson-Crick base pairing is programmed from sequence to fold complex target 2D and 3D geometries[1,2]. In particular, scaffolded DNA origami[3] leverages a long, single-stranded DNA (ssDNA) scaffold to template complementary short staple ssDNA strands to form stoichiometrically well-defined final origami products of programmed dimensions and geometry. Traditionally, 2D[3] and 3D[4,5] bricklike origami objects were designed using parallel duplexes assembled on square and honeycomb lattices largely to endow structural rigidity in 3D, and ease of manual scaffold routing in 2D and 3D, aided by the graphical design program caDNAno[6] that also performs staple assignment semi-automatically. Such 2D and 3D assemblies have been used for a variety of applications[1,2,7–9] including templating materials such as carbon nanotubes[10], metal nanowires[11], nanoparticle coordination[12–15], graphene sheets[16], and cell ligand patterning[2,17]. In addition, functionalized 2D DNA origami has been used as single-molecule chemical reactors[18], energy transfer devices[19], as well as scaffolds for drug delivery[20] and lithographic patterning[21]. These preceding applications rely on the ability to perform secondary functionalization of DNA origami at predefined locations, as well as their structural rigidity. However, the dense packing of DNA duplexes in these 2D and particularly 3D objects limits the overall dimensions that can be achieved using the conventional M13 scaffold used to fabricate DNA origami; high ionic strength conditions are typically required to stabilize bricklike origami due to the high charge density of DNA bundles; and objects with complex boundaries and internal structure are challenging or impossible to design due to the geometric constraints that are imposed by parallel duplexes.

As an alternative, wireframe scaffolded[22–27] and non-scaffolded[28,29] DNA origami design strategies have been introduced to realize complex 2D and 3D geometries without the constraints imposed by densely packed, parallel duplexes. However, the complex scaffold routing and staple design needed to realize these objects requires the use of fully automated sequence design procedures for practical applications. Specifically, vHelix-BSCOR[25] and PERDIX[26] have been introduced for 2D wireframe scaffolded DNA origami, and vHelix-BSCOR[22], DAEDALUS[24], and TALOS[27] for 3D polyhedral origami assemblies. To realize wireframe designs, vHelix-BSCOR employs single-duplex DNA edges, whereas PERDIX renders the target geometry fully automatically in 2D using dual-duplex (DX-based) edges and allows for arbitrary edge lengths, vertex degrees, and vertex angles. However, 2D assemblies programmed using vHelix-BSCOR or PERDIX are limited in their structural fidelity and mechanical stiffness apparent in atomic force microscopy (AFM) and transmission electron microscopy (TEM) imaging due to the use of only two parallel duplexes per edge. This limitation becomes particularly apparent when edge lengths approach the ~50 nm bare persistence length of DNA[30,31]. Similar results of limited structural fidelity are observed from EM in 3D polyhedral assemblies when single[22] or dual-duplex[24] edges are employed, which recently motivated the development of the honeycomb-edge-based approach TALOS to program 3D polyhedra[27].

Initially, the honeycomb approach was used to form the 2D crystallization[32] of origami tiles based on two layers of each origami tile that each has opposite orientations relative to the plane of the tile. More recently, Hong et al.[33] used a multilayered framework to create well-controlled DNA origami wireframe objects, with folded DNA origami showing precise control over corner angles. However, scaffold routing and staple sequence design for these approaches must manually be performed for each target object, and are limited to various patterns of multi-arm

junctions needed for secondary chemical and molecular functionalization at predefined spatial positions.

To enable the generalized design and fabrication of mechanically stiff 2D wireframe DNA origami objects of custom shape[3,12,23,25,26,29,34], here we introduce the fully automatic inverse sequence design procedure METIS (Mechanically Enhanced and Three-layered orIgami Structure) with a simple web interface (https://metis-dna-origami.org). METIS programs lattice-based DNA assemblies by employing three layers corresponding to a cross-section of the six-helix bundle[35] (6HB) and a three-way vertex crossover motif in which every duplex at each layer is connected to another duplex in the same layer in a neighboring wireframe edge, as implemented for 3D DNA origami using TALOS[27]. As in PERDIX[26], unpaired scaffold nucleotides are introduced at vertices to accommodate 5'- and 3'-end misalignments that allow for arbitrary edge lengths and vertex angles to be designed, facilitating the top-down specification of nearly arbitrary 2D shapes and topologies for nanoscale materials science and engineering.

We demonstrate the utility of our automatic design procedure by applying it to fabricate various 2D objects of differing vertex types with and without an internal mesh, and comparing target geometries with corresponding single-layer/DX-based wireframe DNA origami designed using PERDIX[26]. Monodispersity of folded products is confirmed using AFM, and TEM offers quantitative evaluation of the fidelity of programmed internal angles. Molecular dynamics simulations corroborate quantitatively the degree of enhanced mechanical stiffness of honeycomb-based 2D origami designs compared with their DX-based counterparts. METIS should offer a versatile new design framework for fabricating 2D origami objects with enhanced structural rigidity to complement multilayer bricklike assemblies, yet without the geometric limitations imposed by their uniformly parallel duplex, lattice-based design.

## Results

**Automated design of 2D wireframe origami with 6HB edges.** Automatic scaffold routing and staple design for target 2D geometries is based foundationally on the DX-based sequence design approach PERDIX, published previously[26]. Like PERDIX, METIS also utilizes two possible types of geometric input, namely complete line-based rendering of the boundary and internal geometry of the target 2D object versus only border-based rendering of the target object, with the internal geometry determined automatically by the algorithm (Fig. 1a). Importantly, the designed edges for both PERDIX and METIS do not need to correspond to a multiple of a full turn of double helical B-form DNA (10.5-bp)[26,27], allowing for a significantly broader design space compared with other approaches that require discrete edge lengths based on the helicity of B-form DNA[24]. However, in contrast to PERDIX that employs a single-vertex scaffold crossover between each pair of neighboring edges and a single layer corresponding to two parallel DNA duplexes per edge, limiting mechanical properties and structural fidelity (Fig. 1b). To realize honeycomb-based 2D origami assemblies, three discrete layers of dual-duplex edges are employed with antiparallel double-crossovers between neighboring edges (Fig. 1c). The single-stranded scaffold routes the entire target DNA origami object using these three layers, with staple segments connecting the three-layered scaffold path to form the honeycomb 2D wireframe object (Fig. 1d and Supplementary Fig. 1). We introduce a multiway vertex crossover motif[27] in which every duplex within each layer is extended at vertices to covalently connect to its neighboring duplex of the same layer using both scaffold and staple crossings. By introducing the 6HB

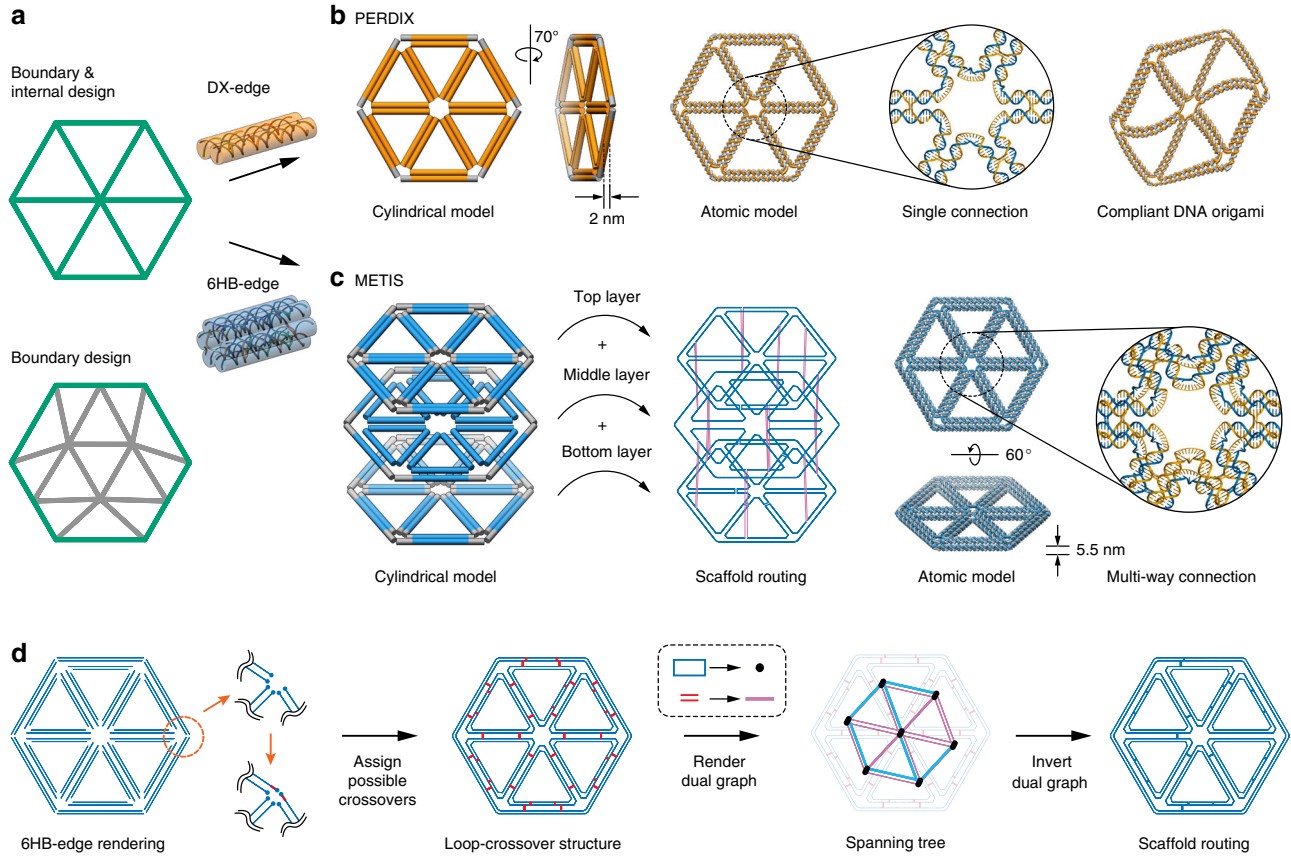

**Fig. 1** Design of 2D wireframe scaffolded DNA origami objects with DX and 6HB edges. **a** Arbitrary target geometries can be specified as input in one of two ways: Boundary and internal design, specifying the complete internal and boundary geometry using piecewise continuous lines; or Boundary design, defining only the border of the target object, with the internal mesh geometry designed automatically. **b** DX-based 2D wireframe scaffolded DNA origami objects published previously, PERDIX[26]. Each wireframe edge is connected covalently to its neighboring edges by one scaffold and one staple crossing. **c** 6HB-based 2D wireframe scaffolded DNA origami, METIS. This 6HB geometry forms three layers connected with scaffold double-crossovers. Each wireframe edge is connected covalently to its neighboring edges by three scaffold and staple crossings. **d** The target geometry presents six DNA duplexes per wireframe edge and forms closed loops with geometrically allowable scaffold double-crossovers between them. The dual graph of the loop-crossover structure is obtained by converting each closed scaffold loop to a node and each possible scaffold double crossover connecting them to an edge. The minimum spanning tree of the dual graph was then determined and inverted, defining the DNA scaffold routing.

edges and the multiway vertex crossover motif, both edge-bending and out-of-plane bending stiffness is enhanced significantly[36] relative to single layer, DX-based 2D wireframe assemblies[12,23,26].

**Design of variable vertex numbers**. Staple sequences designed by METIS following scaffold routing are validated experimentally by one-pot mixing and folding of DNA strands with an annealing temperature ramp and characterizing structural formation using atomic force microscopy (AFM) and transmission electron microscopy (TEM) imaging. We compared the structural fidelity of 6HB-based 2D wireframe origami of different degrees of *N*-arm junctions with DX-based design from PERDIX (Fig. 2). We analyzed three objects including the square, pentagon, and hexagon of 84-bp edge length assigned to the shortest edge (Supplementary Fig. 2 and Supplementary Table 1). AFM indicated that all DX-based objects generally formed well (Fig. 2a–c; left in each panel, Supplementary Figs. 3–5) and are structurally stiffer than single-duplex edge structures[25]. However, the pentagon and hexagon were still relatively flexible compared with the square because more unpaired scaffold and staple nucleotides were introduced to accommodate 5′- and 3′-end misalignments between every two neighboring connected duplexes at the central vertex (Supplementary Table 1). In contrast, 6HB-based DNA

origami objects of variable vertex numbers showed successful formation of designed geometries with vertices of high structural fidelity even when higher order vertex numbers were produced (Fig. 2a–c; right in each panel and Supplementary Figs. 6–8). As expected, the designed internal vertex was precisely controlled using both 6HB edges and multiway connections crossing the vertex, leading to the formation of variable arm junctions of controlled number. To further characterize 6HB-based DNA assemblies, each object was visualized with TEM (Fig. 2d). Wide-field view TEM imaging showed in each case that particles are well folded and monodisperse (Supplementary Figs. 9–12). To test the generality and robustness of our autonomous sequence design procedure, we also applied it to the hexagon with 63-bp edge length. Full-field AFM and TEM imaging of the same folded samples (Supplementary Figs. 13 and 14) revealed high rates of proper formation with precisely controlled internal angles of vertices.

**Precisely controlled vertex angles**. The ability to control angles between edges intersecting at a vertex is an important challenge for the precise design of wireframe origami architectures. To evaluate the ability of METIS to control 2D vertex angles we created two simple objects with a radius of a circumscribed circle of 25 nm—a triangle and a hexagon without any internal mesh

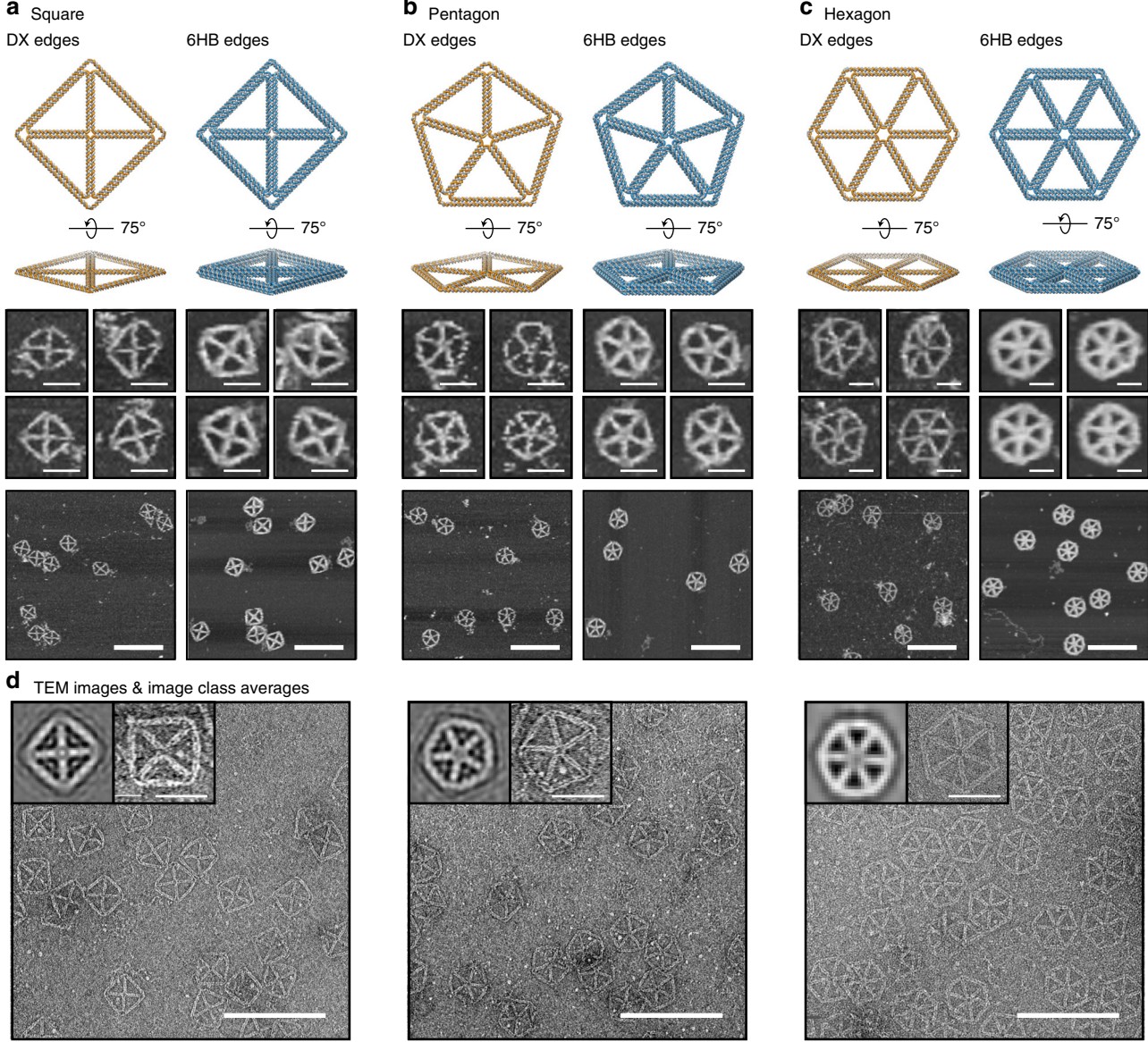

**Fig. 2** Designing variable vertex numbers with DX- and 6HB-based 2D DNA origami objects of 84-bp edge length. The square **a**, pentagon **b**, and hexagon **c** consisting of variable vertex numbers are characterized using AFM. (middle) The top zoomed-in images in each panel are representative objects that are deformed and the two images below these represent objects that are undeformed in shape, where all images are taken from the wide-field view shown beneath. **d** TEM micrographs including zoomed-out and class average images show monodisperse 6HB-based *N*-sided polygons. Scale bars in zoomed-in and zoomed-out AFM and TEM images are 50 and 200 nm, respectively.

structure to reinforce it (Fig. 3a, b, Supplementary Fig. 15, and Supplementary Table 2). For DX-based triangular and hexagonal objects, AFM and TEM imaging confirmed the successful assembly of these structures with high yield (Supplementary Figs. 16–19), but most structures presented rounded and smooth overall shapes with excessive flexibility leading to highly contorted configurations, as may be anticipated due to the bare 50 nm persistence length of DNA that is on the order of the edge lengths themselves. As a consequence, DX-based hexagonal origami objects without internal meshes are structurally unstable and unable to maintain prescribed angles between the two crossing arms at each vertex for the 74-bp edge-length object examined. In contrast, AFM and TEM showed successful formation of the target triangle and hexagon using 6HB edges and multiway connections crossing the vertex, with full-field images revealing accurate angles between arms and a high rate of proper formation (Supplementary Figs. 20–24). We also measured

quantitatively vertex angles from each DNA origami particle in TEM images, with internal angle distributions showing a significantly, three-fold lower standard deviation of the 6HB-based triangle compared with the DX-based triangle, and the DX- versus 6HB-based hexagon exhibiting a similar fold improvement of ~2.5-fold (Fig. 3a, b; right). Lower standard deviations in internal angles observed for triangular versus non-triangular objects are likely attributable to the intrinsically greater mechanical stiffness associated with triangular objects. We further applied our procedure to design and synthesize 6HB-based square and octagon objects with a radius of a circumscribed circle of 25 nm (Fig. 3c, d). TEM showed successful formation of designed target objects at the single-particle level (Supplementary Figs. 25–28) for both structures. We measured internal angles of the 6HB-based square DNA origami from TEM images, showing a slightly lower standard deviation than that of the 6HB-based hexagonal DNA origami. To evaluate the stability of programmed

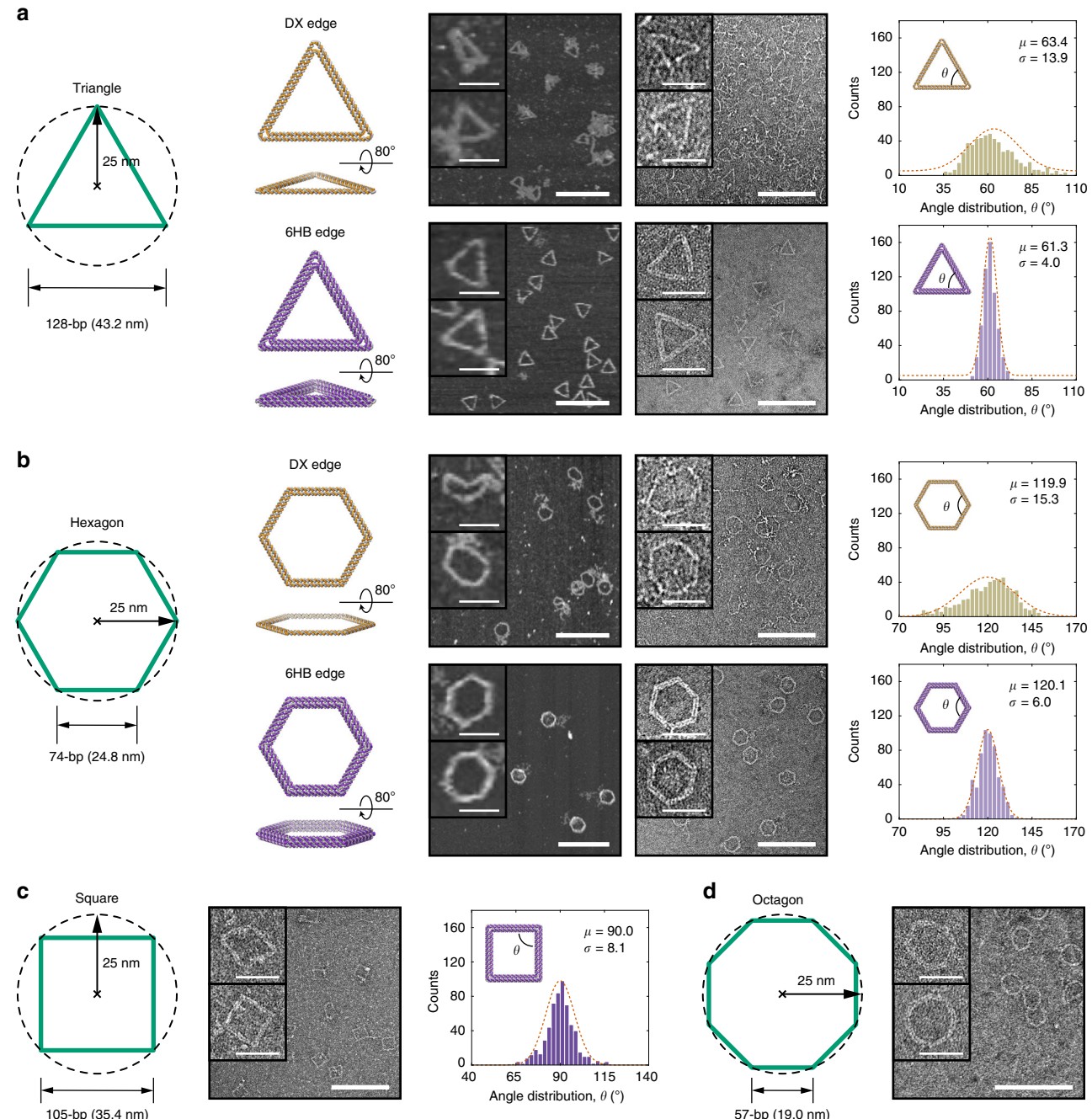

**Fig. 3** Controlled arm angles for a triangle, square, hexagon, and octagon without internal mesh. Target geometries without internal mesh are designed with the radius of the circumscribed circle of 25 nm that corresponds to the triangle **a** with 128-bp edge length and octagon **b** with 74-bp edge length, corresponding to approximately 43 and 25 nm, respectively. Comparison of DX- and 6HB-based DNA origami objects with (left) AFM, (middle) TEM, and (right) histogram analyses of the distributions in angles. **c** TEM and histogram analysis of the distributions in angle of the 6HB-based DNA square of 105-bp edge length. **d** TEM for the 6HB-based DNA octagon of 57-bp edge length. The top zoomed-in image on each panel is the representative object that is the most deformed and the image below represents the object that is the least deformed in shape. Scale bars in zoom-in and zoom-out AFM and TEM images are 50 and 200 nm, respectively. Source Data are available in the Source Data file.

internal vertex angles, we also removed four staples on one edge of the triangle where three scaffold double-crossovers exist (Supplementary Fig. 29). TEM imaging revealed an opening of the now-unconstrained vertices, suggesting that strain may be present at the vertices to force them to open beyond their programmed 60 degrees. We also evaluated the roles of vertex design parameters to examine their impact on vertex angle, choosing 0.42 nm per unpaired nucleotide by default in both the scaffold and staple loop (Supplementary Figs. 29–31).

**Molecular dynamics investigation of structural flexibility**. All-atom molecular dynamics (MD) allowed us to probe the structural flexibility of DX- and 6HB-based triangular DNA origami structures at the edge, base-pair, and atomic level of accuracy. Specifically, four MD simulations were compared, including DX- and 6HB-based versions of a 42- and 84-bp DNA origami triangle (Fig. 4). The distribution of the internal angle from MD is broader for the DX-edge than the 6HB-edge objects, as well as for the 42-bp versus 84-bp objects (Fig. 4b and Supplementary Fig. 32).

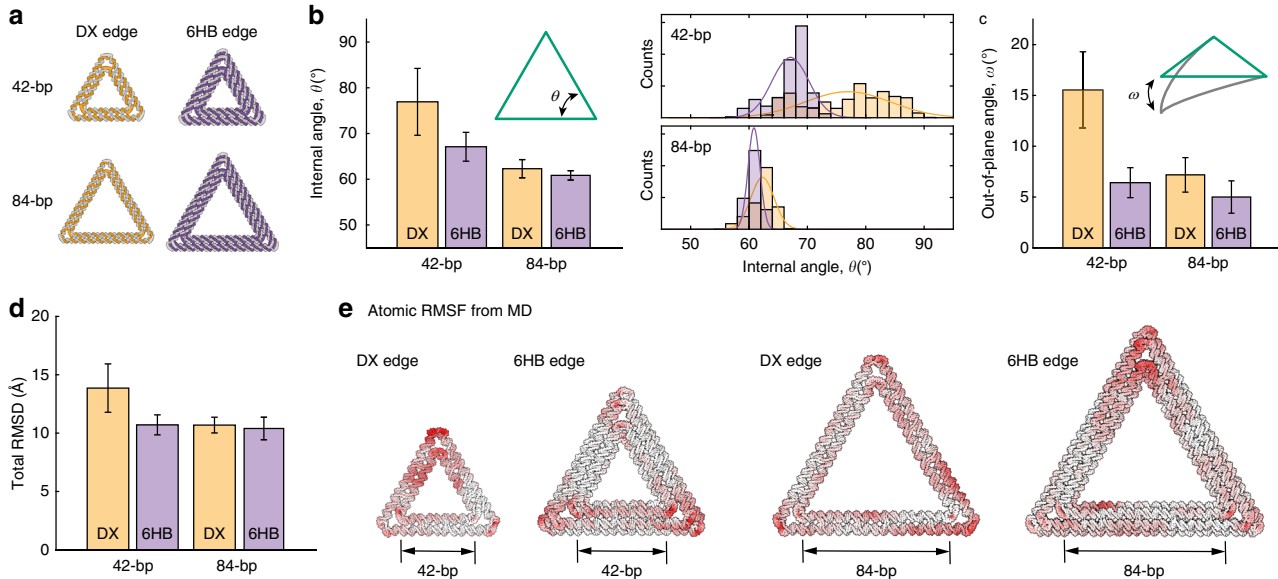

**Fig. 4** Molecular dynamics simulations of triangular DNA origami objects. **a** Two DX- and 6HB-based DNA origami triangular objects are simulated using all-atom molecular dynamics. **b** The average internal angle, $\theta$, of triangular objects varies with edge base-pair length (42- and 84-bp) and edge-type (DX and 6HB edges). **c** The average out-of-plane angle, $\omega$, of the DNA origami triangular objects. **d** Total RMSD of all nucleic acid atoms based on the ground-state atomic model generated by METIS. **e** All-atom RMSFs are calculated for the 42- and 84-bp DX- and 6HB-based triangular objects and mapped as a white-to-red color gradient. Source Data are available in the Source Data file.

Specifically, the internal angle distributions are 76.9 ± 7.3°, 67.1 ± 3.2°, 62.3 ± 2.0°, and 60.8 ± 1.0° for the DX42, 6HB42, DX84, and 6HB84 objects, respectively. Comparison of the former angle distributions (DX-edge vs. 6HB-edge) are also shown experimentally in Fig. 3 for the 128-bp triangular objects, with differences attributable to the increased bending stiffness and persistence length of the 6HB- versus DX-edge objects. Comparison of the latter angle distributions (42-bp vs. 84-bp) is due to the distribution of vertex stress along the shorter edge length, which results in greater instability and bending in the edge (Fig. 4d and Supplementary Fig. 33). In addition, the distributions of out-of-plane bending angles show the same trend (Fig. 4c) as the internal angles, although this is impossible to verify experimentally using the 2D TEM and AFM imaging performed in this study. Similarly, overall thermal flexibility of these triangular objects can be compared using a root-mean-square deviation (RMSD) from the ground-state geometry (Fig. 4d and Supplementary Fig. 32). Specifically, the RMSD distributions are 13.9 ± 2.1, 10.7 ± 0.9, 10.7 ± 0.7, and 10.4 ± 1.0 Å for the DX42, 6HB42, DX84, and 6HB84 objects, respectively. The structural comparisons from the angular distributions are also evident in the total RMSD, where 6HB-based, longer edge-length objects are the least flexible, whereas DX-based, shorter edge-length objects are the most compliant. The sources of local conformational flexibilities are shown from atomic root-mean-square fluctuations (RMSF), which are mapped onto the ground-state structures in Fig. 4e. In all structures, the vertices exhibit the most conformational flexibility, with DX-based objects exhibiting additional flexibility on edges, particularly in the 42-bp structure in which vertex stress can be more easily distributed to the edge. While it would be interesting to simulate the dynamics of larger objects (e.g., 128-bp edge length) in order to test whether these relative conformational trends hold, larger computational resources or coarse-grained models would be required for this purpose[37,38].

Finally, to test the generality and robustness of our design procedure, we used METIS to generate sequence designs and atomic models for ten additional objects (Fig. 5 and Supplementary Table 3) using different mesh types, namely triangular and quadrilateral; with and without internal meshes; and using complex shapes and topology. Two of the designs with quadrilateral meshes were synthesized and characterized experimentally, with AFM and TEM showing high structural fidelity of the target designed objects, and full-field AFM and TEM imaging revealing the successful formation of such objects (Supplementary Figs. 34–37).

## Discussion

Multilayer bricklike DNA origami assemblies are well established to have highest structural fidelity due to their enhanced mechanical stiffness, which also offers high-resolution cryoEM imaging[39]. However, these bricklike assemblies are also limited in the diversity of shapes they can adopt in 2D and 3D compared with wireframe assemblies[24,26]. Here, we introduce a DNA origami wireframe design procedure that combines the structural fidelity of bricklike designs with the geometric versatility of wireframe designs, encoded in the fully automatic, top-down design procedure METIS. We demonstrate geometric versatility of METIS by applying it to a broad variety of highly irregular and asymmetric 2D wireframe geometries, and we validate their structural fidelity quantitatively using TEM and AFM as well as all-atom molecular dynamics compared with conventional DX-based wireframe DNA origami, which exhibit significant edge-bending and distortion of vertex angles. This high structural fidelity of 6HB origami designs is rooted in both their well-known stiff edges but also the highly interconnected vertices consisting of numerous multiway connections, programmed by METIS. This approach now offers the ability to program hollow frame objects with both long, 100 + bp edge lengths without any internal mesh structure. METIS offers various output file formats for use with other design and simulation software including caDNAno[6] files for manual base and oligo editing for functionalization, and Protein Data Bank files[40] for atomic structure visualization and simulation. Theoretically, METIS may be applied to 2D wireframe DNA origami objects with any number of multilayers provided they are of even number, although the complexity of

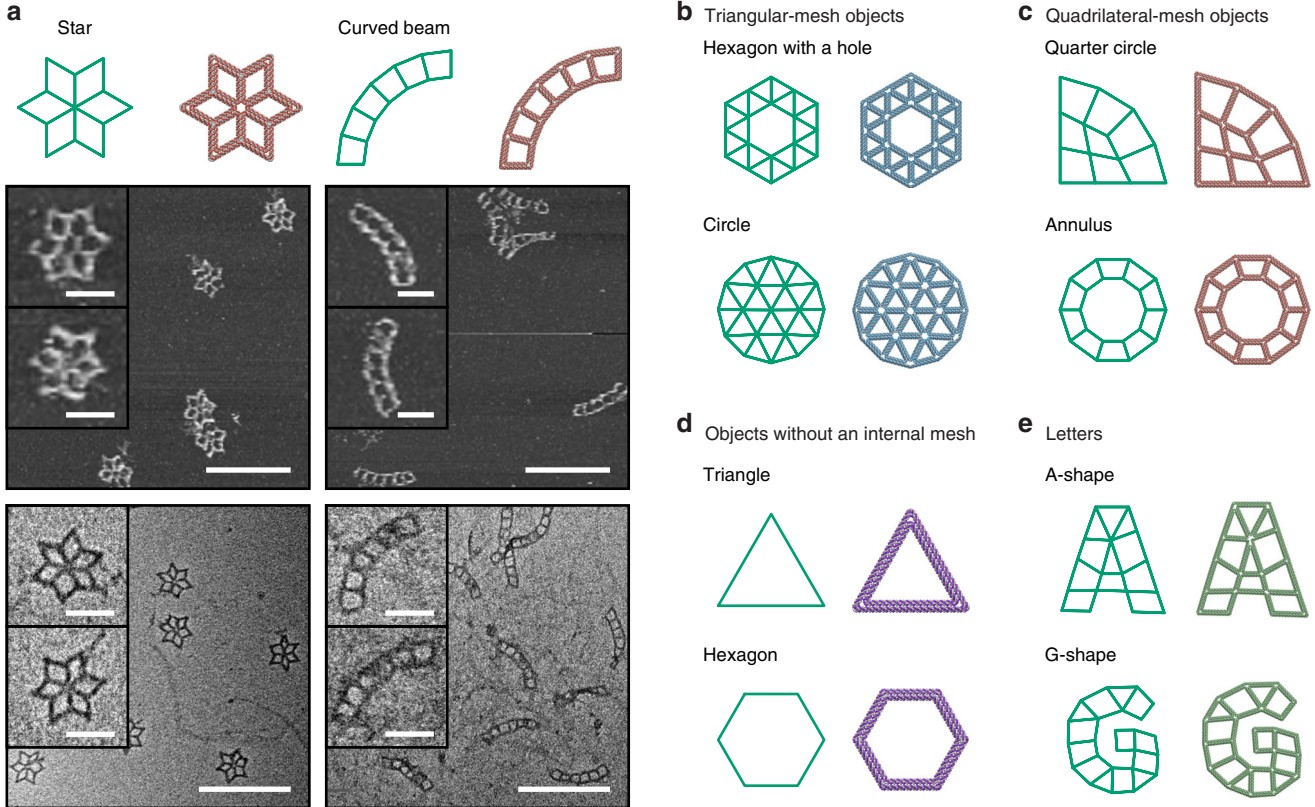

**Fig. 5** Fully automatic sequence design of ten diverse scaffolded DNA origami objects with 6HB edges. **a** Representative (middle) AFM and (bottom) TEM images for a star lattice and curved beam lattice are shown. Target 2D wireframe objects (with green lines) and DNA-based atomic models with **b** triangular and **c** quadrilateral meshes, **d** without internal mesh, and **e** with irregular meshes. The top zoomed-in image on each panel is the representative object that is the most deformed and the image below represents the object that is the least deformed in shape. Scale bars in zoomed-in and zoomed-out AFM and TEM images are 50 and 200 nm, respectively.

staple routing and design for such objects increases significantly over the three-layer, honeycomb case implemented here, and is therefore reserved for future work. Finally, these DNA origami designs should be well-suited to the positioning of bioactive molecules, nanoparticles, and proteins in diverse materials and biomolecular applications that can also be accessed by a broad community of researchers using our open source software and online tool for top-down automated sequence design.

## Methods

**Top-down sequence design**. METIS is provided online for use as standalone open source (https://github.com/lcbb/metis) and a web interface (https://metis-dna-origami.org) for custom design of 2D wireframe scaffolded DNA origami objects. Output files include caDNAno[6] for sequence design editing and oxDNA[37,38] for coarse-grained simulation of structure and conformational dynamics.

**Materials**. DNA origami staple strands were purchased in 96-well plate format from Integrated DNA Technologies, Inc. at 25-nmol synthesis scale, with strands purified by standard desalting and calibrated to 200 μM based on full yield. Staple strands were mixed in equal volume from the corresponding wells and used directly for DNA origami folding without further purification. DNA scaffolds of lengths 2775- and 7249-nt were used (Supplementary Tables 1–4). The 2775-nt DNA scaffold was produced using restriction enzyme cloning. The 2775-nt plasmid assembled using restriction enzyme cloning was transformed into *E. coli* containing the M13cp helper plasmid. The 2775-nt scaffold was subsequently amplified in bacteria in 2xYT incubated for 8 h at 37 °C then harvested and purified[41]. The 7249-nt DNA scaffold (M13mp18) was purchased from Guild BioSciences at a concentration of 100 nM. 10x TAE buffer was purchased from Alfa Aesar. Magnesium acetate tetrahydrate (molecular biology grade) was purchase from MilliporeSigma. 1x TAE buffer with 12.5 mM Mg(OAc)$_2$ was prepared with 10x TAE buffer and Magnesium acetate tetrahydrate. Agarose (molecular biology grade) was purchased from IBI Scientific.

**Origami self-assembly**. All METIS structures were folded with the same protocol. Ten nanomolars of DNA scaffold was mixed with 20 equiv corresponding staples strands (Supplementary Tables 5–20) in 1x TAE buffer with 12.5 mM Mg(OAc)$_2$, the final volume of the self-assembly solution was 50 μl. The mixture buffer solution was annealed in a PCR thermocycler: 95 °C for 2 min, 70–45 °C at a rate of 0.5 °C per 20 min, and 45–20 °C at a rate of 0.5 °C per 10 min. The annealed solution was validated by 1.5% Agarose gel in 1x TAE buffer with 12.5 mM Mg (OAc)$_2$ and 1x SybrSafe. Gels were run at 60 V and subsequently imaged under blue light, or using Typhoon imager (FLA 7000). The annealed solution was diluted into 300 μl with 1x TAE buffer with 12.5 mM Mg(OAc)$_2$, and the extra staple strands were removed with MWCO = 100 kDa spin filter concentration columns. The purified DNA origami solution was adjusted to desired concentrations (5 nM) for AFM and TEM imaging. All DX-based DNA origami objects were folded by mixing 5 nM of its corresponding ssDNA scaffold with a 20 times molar excess of staple strands in 1x TAE buffer with 12.5 mM Mg(OAc)$_2$. The final volume of the mixture was 100 μl. The mixture buffer solution was annealed in a PCR thermocycler from 90 to 4 °C in about 12 h: 90 to 86 °C at a rate of 4 °C per 5 min, 85 to 70 °C at a rate of 1 °C per 5 min, 70 to 40 °C at a rate of 1 °C per 15 min, and 40 to 25 °C at 1 °C per 10 min, held at 4 °C in the end[26].

**AFM and TEM imaging**. AFM imaging was performed in "ScanAsyst mode in fluid" (Veeco Multimode 8) with ScanAsyst-Fluid + or SNL-10 tips (Bruker Inc.). Two microliters of sample (5 nM) were deposited onto freshly cleaved mica (Ted Pella Inc.), and 0.5–1.0 μl of NiCl$_2$ at a concentration of 100 mM were added to the samples to fix the origami nanostructures on the mica surface. After waiting for ~30 s for sample adsorption to mica, 80 μl of 1x TAE/Mg$^{2+}$ buffer was added to the samples, and an extra 40 μl of the same buffer was deposited onto the AFM tip. For TEM imaging, 5 μL of DNA origami solution (5 nM) was deposited onto fresh glow discharged carbon film with copper grids (CF200H-CU; Electron Microscopy Sciences Inc., Hatfeld, PA), and the sample was then allowed to absorb onto the surface for 30 s. After the sample solution was blotted from the grid using Whatman 42 filter paper, the grid was placed on 5 μL of freshly prepared 2% uranyl-formate with 25 mM NaOH for 10 s. The remaining stain solution on the grid was blotted away using Whatman 42 filter paper and dried under house vacuum prior to imaging. The sample was imaged on a Technai FEI with a Gatan

camera. 2D class averages were generated with EMAN2 from negative stained TEM images.

**All-atom molecular dynamics (MD) simulations**. All-atom MD simulations were performed for 42- and 84-bp triangular wireframe DNA origami objects to compare the stability and dynamical motion of the DX- vs. 6HB-based design. The PDB files for the initial atomic coordinates of the DX- and 6HB-based 2D wireframe DNA origami were generated using PERDIX[26] and METIS, respectively. All-atom DNA objects were solvated in TIP3P water[42] with explicit $Mg^{2+}$ and $Cl^-$ ions added to neutralize DNA charges and to set the ion concentration to 12 mM, which is consistent with the experimental conditions. MD simulations were performed with the program NAMD2[43] using the CHARMM36 force field[44–47] and Allnér $Mg^{2+}$ parameters[48]. During MD simulations, an integration time step of 2 fs was used with periodic boundary conditions applied to an orthogonal simulation cell, and van der Waals energies calculated with a 12 Å cut-off and switching function applied from 10 to 12 Å, and a 14 Å pair list distance. The Particle Mesh Ewald (PME) method[49] was used to calculate full electrostatics with a maximum grid point spacing of 1 Å. Full electrostatic forces were computed every two time steps (every 4 fs) and non-bonded forces were calculated at each time step (2 fs). Equilibration and production simulations were performed in the *NpT* ensemble using the Nosé-Hoover Langevin piston method[50,51] for pressure control with an oscillation period of 200 fs and a damping time of 100 fs. Langevin forces were applied to all heavy atoms for temperature control (300 K) with coupling coefficients of $5\,ps^{-1}$. All hydrogen atoms were constrained to their equilibrium lengths during the simulations and atomic coordinates were recorded every 1 ps for downstream analysis of coordinate trajectories. Prior to production MD, solvent and ions were allowed to equilibrate for 1 ns while the nucleic acid atoms were spatially constrained. For production MD, the DNA nanostructures were run for a total of 200 ns each. While 50 ns equilibration did not result in full equilibration of all of the origami objects, this equilibration time was chosen as a compromise to allow for a reasonable total computation time.

**Structural analysis of MD trajectories**. Atomic coordinates for the triangular wireframe DNA origami objects (42- and 84-bp edge length, DX- and 6HB-based object) were extracted from the production MD simulations every 1 ns. At each sampled time point, atomic coordinates were superposed onto the reference geometric coordinates ($t = 0$ ns), and the root-mean-square deviation (RMSD) of the sampled atomic coordinates was calculated with respect to these reference coordinates. The total RMSD of all atoms was calculated at each time point. After equilibration time of ~50 ns, the total RMSD was calculated from the average of three separate bins of 50 ns (50–100 ns, 100–150 ns, and 150–200 ns). Superposition of atomic coordinates and RMSD calculations were performed using the Python package ProDy[52]. Also, root-mean squared fluctuations (RMSF) of all atomic coordinates were averaged over the production MD simulations for each triangular wireframe structure, again performed using the Python package ProDy[52]. To determine the average internal angle, $\theta$, of each triangular wireframe object, a geometric approach for analyzing each frame in the MD trajectory was utilized. At each vertex, two *M*-bp DNA *N*-helix edges are connected, and the bp of each edge are indexed as $bp_{1,1}, ..., bp_{1,M}, ..., bp_{N,1}, ..., bp_{N,M}$. Using the ProDy[52] package, the geometric center of atoms in each bp ($x_{1,1}, ..., x_{N,N}$), and the geometric center of atoms in each pair of bp ($bp_{1,i}, bp_{2,i}$) for DX- or sextet of bp for 6HB-based object, $i = 1, 2, ..., M$, denoted $c_i$, is calculated at each MD frame. A right-handed orthonormal basis ($b_1, b_2, b_3$) is defined using the three principal axes of the point cloud $\{c_1, ..., c_M\}$, in which $b_1$ is coincident with the first principal axis and points from $c_1$ to $c_M$, $b_2$ is coincident with the second principal axis and points from the inner to the outer bp, and $b_3$ is coincident with the third principal axis and points outward of the nanoparticle. Next, a *L*-bp region at the starting end of each edge is selected to define a vector $e_1$, which is coincident with the first principal axis of the point clouds $\{c_1, ..., c_L\}$, and points from $c_1$ to $c_L$. For the 42-bp and 84-bp objects, $L$ was chosen as 20 and 40 bps, respectively. At each frame, a triangular plane is determined from the three center points of the edges ($c_{M/2}$), and the normal to this plane is calculated as $c_{norm}$. Next, both left- and right-hand edge vectors of a triangular vertex are projected onto the $c_{norm}$ plane as $e_{l,proj}$ and $e_{r,proj}$, respectively. The internal angles at each vertex are then $\theta_i = e_{l,proj} \cdot e_{r,proj}$, where $i = 1, 2, 3$, and the average internal angle at each frame is $\theta$. In addition, the out-of-plane angles at each vertex are calculated as $\omega_i = e_{norm} \cdot c_{norm}$, where $i = 1, 2, 3$, $e_{norm} = e_1 \times e_2$, and the average out-of-plane angle at each frame is $\omega$.

## Code availability

Computer code is available from GitHub at https://github.com/lcbb/metis.

## Data availability

Representative image data generated during and/or analyzed during the current study are shown in Figures and Supplementary Materials. Source Data are available in the Source Data file. The datasets generated during and analyzed the current study are available from the corresponding author on reasonable request.

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

## Acknowledgements

Funding from the National Science Foundation CCF-1564025, CBET-1729397, and CHE-1839155, and the Office of Naval Research N00014-17-1-2609 are gratefully acknowledged. Research was sponsored by the U.S. Army Research Office and was accomplished under Cooperative Agreement Number W911NF-19-2-0026 for the Institute for Collaborative Biotechnologies. T. Shepherd and H. Huang are acknowledged for technical assistance in producing the 2275-nt DNA scaffold.

## Author contributions

H.J., X.W., and M.B. conceived of the sequence design approach based on the 6HB-based DNA 2D origami object with multiway connections. H.J. implemented the design algorithm and web portal, and processed the results to make the figures. X.W. implemented the experimental assay and collected the experimental data. H.J. and X.W. measured angles from TEM imaging and analyzed the data. W.P.B. performed the molecular dynamics simulations and analyzed the trajectories. M.B. supervised the project. All authors discussed the results and wrote and edited the paper.

## Competing interests

The authors declare no competing interests.
