## [Peer Review File · Nature Communications]

Reviewers' Comments:

Reviewer #1:

Remarks to the Author:

This study develops a new design tool for structural DNA nanotechnology. Simplifying and automating the design process of DNA nanostructures, and in particular scaffolded DNA origami structures, has been an active and important area of research in the last several years. There have been a few important contributions including by the authors' group. This study makes a key advance over these prior studies to expand the limits of automated structure design and improve structural stability. It appears to me the proposed method also provides generally improved folding quality and uniformity relative to prior automated design methods, likely because of the improved stiffness and structural stability but I believe this is still an important point. This study particularly focuses on plan wireframe structures, developing a new approach to design structures (and automate the sequence design process) following a three layer approach that enables flexible design of vertices, and hence design of a wide range of 2D wireframe geometries. Wireframe structures have found many applications, so I think automating the design of high quality wireframe structures is likely to be of broad interest to researchers in DNA nanotechnology and many other fields. Overall this is an impressive study that develops a useful tool. I have some questions, minor comments, and suggestions below.

- "which recently motivated the development of the honeycomb-edge-based approach TALOS to program 3D polyhedra.", seems like reference is missing at end of this sentence.
- Folding of 84bp edge length structures is impressively uniform in both AFM and TEM images. Nearly every structure appears nicely folded with crisp edges, which is often not the case for wireframe structures and highlights the benefit of the stiffer and thicker bundles. The 63bp edge structures do not appear to form uniformly as well (Supp fig 13). While the folding is still quite good, it does not appear as clean as the 84bp edge length. Is there any limitation with this edge length, or perhaps this is simply an imaging artifact, AFM might reveal the 63bp structures are similarly nicely folded.
- As a general question for the supplemental figures, are the top and bottom TEM images just two examples from the same batch of structures? I would assume so, which is fine, but if not it would be worth mentioning as that provides another metric of repeatable high quality folding. Also, are those gel purified structures for both TEM and AFM images?
- Angle distributions in figure 3 appear to use sigma for the average and mu for the standard deviation. That is confusing and I would suggest switching those to follow more standard use with mu as average and sigma as standard deviation
- The shape fidelity for the octagon appears to be not as good as the other structures. The hexagon seems remarkably uniform with sharp vertices, whereas some of the octagon vertices appear more round. The fact that more structures form open loops would also suggest there is more strain in the closed structures. Although the folding results are still quite good, this may give insight into the vertex/angle design, perhaps there are limitations for shallow angles as in the octagon?
- It is interesting that the longer edge lengths result in stiffer structures. Is this purely because of the vertex stress? Are the vertex designs identical? If so my intuition would suggest that the shorter structures would be stiffer. Is this simply due to the localization of stresses? Or can the authors comment on why this is the case?
- The authors should mention the web interface in the main body of the manuscript in addition to the methods. That is a useful point to make it broadly usable by researchers, and currently it is somewhat buried in the methods.

Reviewer #2:

Remarks to the Author:

This manuscript describes the application of methods developed for automated computational

design of DNA origami, to enable the fabrication of mechanically stiff 2D wireframe DNA origami objects of custom shape. The authors introduce the fully automatic, top-down design procedure called METIS. METIS is now provided online for use as open source and a web interface is already available. They demonstrate the method using several simple 2D DNA origami samples. However, while idea to develop these approaches is not completely novel, PERDRIX or TALOS have been introduced for 2D wireframe scaffolded DNA origami or to program 3D polyhedra, this is a more ambitious use of the technique. The study performed here is well suited to Nature Communications, as it contains a careful description of the method itself along with some guidance for others wishing to apply the method in their own work.

The manuscript is clear, the experimental and theoretical studies all seemed to be competently executed, and sufficient detail seems to be present for replicating the work. The applicability of the method seems reasonably broad, however, the illustration still lacks the evidence of its applicability to DNA origami structures with higher number of multilayers.

Having said that, this remains a very solid study, and in my opinion is of sufficiently broad interest to justify publication in Nature Communications. My only very minor suggestions are listed below. Otherwise I find the manuscript suitable for publication largely as-is.

#1. The provided figure 1 would benefit from a more detailed schematic drawing, especially about the wireframe edge design.

#2. Supplementary Figure 23. The lane labels appear to be switched.

#3. Paragraph starting with " Finally, METIS offers various output formats for use with other DNA structural design software including although the staple routing and design for such objects increases significantly over the three-layer, honeycomb case, and is therefore reserved for future work." This paragraph can be moved from Introduction to Discussion.

Reviewer: 1

This study develops a new design tool for structural DNA nanotechnology. Simplifying and automating the design process of DNA nanostructures, and in particular scaffolded DNA origami structures, has been an active and important area of research in the last several years. There have been a few important contributions including by the authors' group. This study makes a key advance over these prior studies to expand the limits of automated structure design and improve structural stability. It appears to me the proposed method also provides generally improved folding quality and uniformity relative to prior automated design methods, likely because of the improved stiffness and structural stability but I believe this is still an important point. This study particularly focuses on plan wireframe structures, developing a new approach to design structures (and automate the sequence design process) following a three layer approach that enables flexible design of vertices, and hence design of a wide range of 2D wireframe geometries. Wireframe structures have found many applications, so I think automating the design of high quality wireframe structures is likely to be of broad interest to researchers in DNA nanotechnology and many other fields. Overall this is an impressive study that develops a useful tool. I have some questions, minor comments, and suggestions below.

We highly appreciate the positive assessment of our work by the Reviewer.

(1) "which recently motivated the development of the honeycomb-edge-based approach TALOS to program 3D polyhedra.", seems like reference is missing at end of this sentence.

Thank you for pointing out this omission, which has been corrected in our revised manuscript to reference (Jun et al, *ACS Nano* (2019)).

Main Text, Page 2, Lines 15–17

which recently motivated the development of the honeycomb-edge-based approach TALOS to program 3D polyhedra²⁷.

(2) Folding of 84bp edge length structures is impressively uniform in both AFM and TEM images. Nearly every structure appears nicely folded with crisp edges, which is often not the case for wireframe structures and highlights the benefit of the stiffer and thicker bundles. The 63bp edge structures do not appear to form uniformly as well (Supp fig 13). While the folding is still quite good, it does not appear as clean as the 84bp edge length. Is there any limitation with this edge length, or perhaps this is simply an imaging artifact, AFM might reveal the 63bp structures are similarly nicely folded.

The 63bp edge length structures have also been characterized using AFM. The structures are well folded and appear to be uniform, comparable with the 84bp structures. However, because there is additional unpaired scaffold present in the former constructs, parts of the images are often blurred under AFM. For TEM, we suspect there may also be artifacts present due to unpaired scaffold. We also appreciate and agree that different edge lengths may impact structural integrity in some cases, although we have no clear evidence of this at this stage. We have added the new AFM imaging results to Supplementary Figure 13.

Supplementary Figure 13 | AFM imaging of 6HB-based hexagonal DNA origami of 63-bp edge-length.

(3) As a general question for the supplemental figures, are the top and bottom TEM images just two examples from the same batch of structures? I would assume so, which is fine, but if not it would be worth mentioning as that provides another metric of repeatable high quality folding. Also, are those gel purified structures for both TEM and AFM images?

Yes, the top and bottom TEM images are two examples from the same batch of structures.

We use MWCO = 100 kDa spin filter purified samples for both TEM and AFM images, which we note in Material and Methods.

We appreciate the suggestion of the reviewer and have added the following statement to further corroborate high quality folding of the structures: “High quality folding is additionally confirmed by TEM and AFM imaging of the same folded batch of origami objects...”

(4) Angle distributions in figure 3 appear to use sigma for the average and mu for the standard deviation. That is confusing and I would suggest switching those to follow more standard use with mu as average and sigma as standard deviation

We appreciate the Reviewer pointing this error out, which has been corrected in the revised manuscript.

Figure 3 | Controlled arm angles for a triangle, square, hexagon, and octagon without internal mesh.

(5) The shape fidelity for the octagon appears to be not as good as the other structures. The hexagon seems remarkably uniform with sharp vertices, whereas some of the octagon vertices appear more round. The fact that more structures form open loops would also suggest there is more strain in the closed structures. Although the folding results are still quite good, this may give

insight into the vertex/angle design, perhaps there are limitations for shallow angles as in the octagon?

The opening of the octagonal structure is indeed interesting. For the octagonal DNA origami without any internal mesh in Figure 4d, there are three scaffold crossovers located in the same edge to form a “seam” (Design Type 1, see Figure below), which possibly results in the opening of the edge of this structure. To test this hypothesis, we modified the scaffold routing by moving scaffold double-crossovers to different edges (Design Type 2). Interestingly, the folding yield of this new octagonal structure improved, as observed using AFM, TEM, and gel electrophoresis. However, the shape of the octagon still appears round because of both its shallow angles and its short edge lengths compared with the other objects.

We have updated results for the new octagonal structure in Figure 3 and Supplementary Figures 26 and 28.

We have also updated the online algorithm and software METIS to generate objects in this manner, namely to eliminate double-crossover seams in edges, when they do not have an internal mesh.

Supplementary Figure 26 | Two designs, agarose gel electrophoresis, and AFM imaging of 6HB-based octagonal DNA origami of 57-bp edge-length without internal mesh. All scaffold double-crossovers are located on the same edge in Design Type 1 versus distributed to different edges in Design Type 2.

Supplementary Figure 28 | TEM imaging of 6HB-based octagonal DNA origami of 57-bp edge-length without internal mesh when using Design Type 2 shown in Supplementary Figure 26.

(6) It is interesting that the longer edge lengths result in stiffer structures. Is this purely because of the vertex stress? Are the vertex designs identical? If so my intuition would suggest that the shorter structures would be stiffer. Is this simply due to the localization of stresses? Or can the authors comment on why this is the case?

The routing pattern of the vertex generally depends on the vertex angle and number of incoming edges of the target geometry. In the case of the same geometry, the routing pattern of the vertex design also depends on the edge length.

While the effect of edge length on structural integrity is an interesting topic, thus far we do not have clear evidence demonstrating that longer edge lengths result in stiffer structures, which would indeed be non-intuitive. While our results show that the 128-bp triangle DNA origami has higher structural integrity and than other objects with greater numbers of edges and shorter edge-length, this is very likely due to the intrinsic mechanical stiffness associated with triangles generally, and equilateral triangles in particular, rather than the edge-length per se. To address this point, we have added the following sentence to Results:

Main Text, Page 4, Lines 20–22

Lower standard deviations in internal angles observed for triangular versus non-triangular objects are likely attributable to the intrinsically greater mechanical stiffness associated with triangular objects.

(7) The authors should mention the web interface in the main body of the manuscript in addition to the methods. That is a useful point to make it broadly usable by researchers, and currently it is somewhat buried in the methods.

We appreciate this point and have included mention of this in the main body of the revised manuscript.

Main Text, Page 2, Lines 28–30

here we introduce the fully automatic inverse sequence design procedure METIS (Mechanically Enhanced and Three-layered origami Structure) with a simple web interface (<https://metis-dna-origami.org>).

Reviewer: 2

This manuscript describes the application of methods developed for automated computational design of DNA origami, to enable the fabrication of mechanically stiff 2D wireframe DNA origami objects of custom shape. The authors introduce the fully automatic, top-down design procedure called METIS. METIS is now provided online for use as open source and a web interface is already available. They demonstrate the method using several simple 2D DNA origami samples. However, while idea to develop these approaches is not completely novel, PERDRIX or TALOS have been introduced for 2D wireframe scaffolded DNA origami or to program 3D polyhedra, this is a more ambitious use of the technique. The study performed here is well suited to Nature Communications, as it contains a careful description of the method itself along with some guidance for others wishing to apply the method in their own work.

The manuscript is clear, the experimental and theoretical studies all seemed to be competently executed, and sufficient detail seems to be present for replicating the work. The applicability of the method seems reasonably broad, however, the illustration still lacks the evidence of its applicability to DNA origami structures with higher number of multilayers.

Having said that, this remains a very solid study, and in my opinion is of sufficiently broad interest to justify publication in Nature Communications. My only very minor suggestions are listed below. Otherwise I find the manuscript suitable for publication largely as-is.

We greatly appreciate the positive assessment of our work by the Reviewer.

(1) The provided figure 1 would benefit from a more detailed schematic drawing, especially about the wireframe edge design.

We appreciate this point and have added detailed schematic drawing in the revised Figure 1.

Figure 1 | Design of 2D wireframe scaffolded DNA origami objects with DX and 6HB edges.
(a) Arbitrary target geometries can be specified as input in one of two ways: Boundary and internal design, specifying the complete internal and boundary geometry using piecewise continuous lines; or Boundary design, defining only the border of the target object, with the internal mesh geometry designed automatically. (b) DX-based 2D wireframe scaffolded DNA origami objects published previously, PERDIX²⁶. Each wireframe edge is connected covalently to its neighboring edges by one scaffold and one staple crossing. (c) 6HB-based 2D wireframe scaffolded DNA origami, METIS. This 6HB geometry forms three layers connected with scaffold double-crossovers. Each wireframe edge is connected covalently to its neighboring edges by three scaffold and staple crossings. (d) The target geometry presents six DNA duplexes per wireframe edge and forms closed loops with geometrically allowable scaffold double crossovers between them. The dual graph of the loop-crossover structure is obtained by converting each closed scaffold loop to a node and each possible scaffold double crossover connecting them to an edge. The minimum spanning tree of the dual graph is then determined and inverted, defining the DNA scaffold routing.

(2) Supplementary Figure 23. The lane labels appear to be switched.

We thank the Reviewer for carefully reviewing our figures, and we have corrected this error in our revised SI.

Supplementary Figure 23 | Agarose gel electrophoresis for 6HB-based triangular and hexagonal DNA origami objects without internal mesh.

(3) Paragraph starting with “ Finally, METIS offers various output formats for use with other DNA structural design software including although the staple routing and design for such objects increases significantly over the three-layer, honeycomb case, and is therefore reserved for future work.” This paragraph can be moved from Introduction to Discussion.

We appreciate this point and have moved this paragraph from Introduction to Discussion accordingly.

Main Text, Page 5, Lines 38–44

METIS offers various output file formats for use with other design and simulation software including caDNAno⁶ files for manual base and oligo editing for functionalization, and Protein Data Bank files⁴⁰ for atomic structure visualization and simulation. Theoretically, METIS may be applied to 2D wireframe DNA origami objects with any number of multilayers provided they are of even number, although the complexity of staple routing and design for such objects increases significantly over the three-layer, honeycomb case implemented here, and is therefore reserved for future work. Finally,...

Reviewers' Comments:

Reviewer #1:

Remarks to the Author:

The authors have addressed all may prior comments appropriately.

Reviewer #1 (Remarks to the Author):

The authors have addressed all may prior comments appropriately.

We greatly appreciate the positive assessment of our work by the Reviewer.